# Food and Nutrition in Autistic Adults: Knowledge Gaps and Future Perspectives

**DOI:** 10.3390/nu17091456

**Published:** 2025-04-26

**Authors:** Sara Remón, Ana Ferrer-Mairal, Teresa Sanclemente

**Affiliations:** 1Departamento de Producción Animal y Ciencia de los Alimentos, Instituto Agroalimentario de Aragón (IA2), Universidad de Zaragoza-CITA, M Servet 177, 50013 Zaragoza, Spain; remon@unizar.es (S.R.); ferrerma@unizar.es (A.F.-M.); 2Facultad de Ciencias de la Salud y del Deporte, Universidad de Zaragoza, Pl. Universidad, 3, 22002 Huesca, Spain

**Keywords:** autism spectrum disorder (ASD), adults, food selectivity, eating behaviors, nutritional status, obesity, nutritional deficiencies

## Abstract

Proper nutrition is a critical component in supporting the overall health and development of individuals with autism spectrum disorder (ASD) who experience eating difficulties associated with their autistic traits. Evidence regarding the prevalence, origins, and consequences of eating issues related to ASD is largely derived from studies on autistic children, while information pertaining to adults remains scarce. It is therefore essential to critically review existing research focusing on autistic adults to draw robust conclusions and identify clear research gaps. A computer-aided search in PubMed, Science Direct, Scopus, and Web of Science databases spanning the years 2013–2024 using the search terms covering ASD/Autism, Adult, Nutrition/Nutritional Status, and Diet yielded 43 full-text articles. In our literature review, we explored three critical aspects of nutrition in adults with ASD: their food preferences and sensory processing patterns, studies on nutritional status, and whether dietary and nutritional interventions have improved their adherence to healthier diets. Autistic adults appear to select food based on sensory perceptions. This selection pattern can affect their nutritional status, with a tendency toward overweight and nutritional deficiencies. The most promising intervention strategies incorporate sensory adaptation and structured meal planning. Further research should apply rigorous methodologies that account for this population’s specific characteristics.

## 1. Introduction

Autism spectrum disorder (ASD) is a heterogeneous set of broad-spectrum neurodevelopmental disorders that includes impairments in social interaction, language, communication, and imaginative play. It also includes restricted, repetitive, and stereotyped patterns of behavior, activities, and interests [1]. In the last two decades, the global prevalence of ASD has undergone a significant increase, associated with changes in diagnostic criteria, greater scientific evidence, and increased familiarity with the disorder on the part of the general population [2,3]. A suggested 1–2% of the general population are autistic [4], although epidemiological figures vary widely on a worldwide scale [5], countries/regions with high income display the highest prevalence [6]. The median male-to-female ratio is 4:2 or even lower in cases of autism with intellectual disability (ID) [5]; this refutes past stereotypes which assumed that autistic people were overwhelmingly men and boys, and only very rarely women and girls.

The core features of ASD often co-occur with a range of physical and psychiatric challenges such as gastrointestinal disorders (constipation, diarrhea, reflux, abdominal bloating, pain, and discomfort), attention-deficit/hyperactivity disorder (ADHD), depressive and anxiety disorders, or eating/feeding problems [3]. Furthermore, adults with ASD have a higher incidence of physical and mental health comorbidities than the general population [7,8]. Autistic adults are at a greater risk of developing diabetes, dyslipidemia, and heart disease [9,10] along with a higher prevalence of overweight/obesity [11] than non-autistic adults. Such risks are accentuated in ASD patients with ID [12] and seniors [13,14,15]. A healthy diet and regular physical activity are determinant in dealing with these highly prevalent comorbidities.

Certainly, proper nutrition is a critical factor in supporting the overall health and development of individuals with ASD. However, adherence to healthy diets is conditioned by several factors, including the fact that many people with ASD have eating issues, such as extreme selectiveness about food, refusal to eat, food neophobia, eating too much or too little, emotional over- or under-eating, and behavioral problems during meals. These behaviors are thought to stem from cognitive and behavioral rigidity, restricted interests, and sensory sensitivities characteristic of ASD. Although the heterogeneity among individuals is important, diets of children and adolescents with ASD often differ compared to their non-autistic peers [16,17,18] Consequently, some nutritional concerns have been identified such as weight problems [19], and various nutritional deficiencies [19,20,21,22,23]. As a result, diets with nutritional supplements or even the elimination of certain nutrients have been prescribed, although such measures do not seem to be supported by clear scientific evidence [22,24,25,26].

However, we should not assume that these conclusions derived from studies with children apply to autistic adults, as their nutritional needs and eating behaviors likely differ significantly. The support needs of people with ASD vary from individual to individual and within the same individual over time, as this condition should be regarded as a set of highly individualized behaviors. There seems to be a consensus that autistic adults are not merely “children writ large” because, in the transition to adulthood, symptoms and behaviors of ASD tend to abate in severity [27] making it an ideal time for learning and reinforcement, especially in terms of nutrition [28]. This has mainly been observed in cognitive-able individuals who have developed ways to circumvent or disguise some of their difficulties [29]. Follow-up studies covering an extended period from childhood to adulthood generally indicate that many individuals show marked improvements in social and communication skills as they grow older [4]. However, the median percentage of ASD cases with co-occurring ID is 33.0% [5]. This issue is crucial: it is not yet clear whether autistic adults with and without ID should be treated differently in this respect, as it is still difficult to determine which portion of present problems is linked to ASD and which part is linked to ID [30]. Similar unknowns arise when we refer to senior citizens on the autism spectrum, a subpopulation currently surrounded by many enigmas [31]. Therefore, understanding the specific nutritional challenges and needs of autistic adults across different life stages is essential to develop appropriate interventions that effectively address their unique requirements.

Scientific publications on food, nutrition, and ASD are attracting attention and increasing in number. However, despite the increase in general publications, those relating to adults are scarce or practically non-existent [32]. In this context, the Erasmus + APIA project (2023-1-FR01-KA220-ADU-000159952) was launched to contribute to the inclusion of adults with ASD in society and to promote their autonomy and well-being through nutrition. APIA aims to develop tools that enable professionals to improve the independence and well-being of adults with ASD through proper nutrition. To draw up these dietary guidelines, it is essential to compile a review of existing combined knowledge on food, diet, and health in autistic adults. Such a review would also help us detect gaps in knowledge, leading to guidelines for future research and the development of specific, personalized nutritional intervention programs that address the particular needs of autistic adults.

## 2. Methodology

This exploratory literature review examines the scientific evidence related to three critical aspects of nutrition in adults with ASD: what we know about their food preferences and sensory processing, studies on nutritional status, and finally, whether dietary and nutritional intervention studies have been conducted to improve adherence to healthier diets. We searched the electronic databases Science Direct, PubMed, Scopus, Web of Science, and Google Scholar to find studies published between 2013 and May 2024 in English, Spanish, French, and Italian. A comprehensive search was performed using the following terms: autism, autistic disorder, ASD, autism spectrum conditions, adult/s, nutrition, diet, food. All related studies were identified and transferred into the Mendeley Reference Manager (web version accessed via https://www.mendeley.com) to select and manage the references. Our only obligatory inclusion criterion was that the study included an adult population with ASD. Non-original research publications were excluded, including review articles, commentaries, editorials, and opinion pieces. To find further potentially eligible studies, we manually screened the reference lists of selected studies along with their citations.

We selected 43 articles focusing on adults with ASD for inclusion in this review. The studies are presented in Table 1, grouped under the following headings: food selectivity, food sensory processing, anthropometric measurements, biochemical, clinical and dietary assessment of a nutritional status, nutritional supplementation and restricted diets, and dietary intervention. As the number of studies focusing on autistic adults is limited, we considered a broad range of candidates, including studies on individuals with a confirmed ASD diagnosis at any time in their life, individuals self-reporting a clinical diagnosis from a healthcare professional, and individuals reporting high autistic traits (without a formal clinical diagnosis). To clarify the generalizability of findings from the wide variety of inclusion criteria used in adults with ASD research, each study’s sample is specified in Table 1.

## 3. Food Selectivity and Sensory Food Processing in Adults with Autism Spectrum Disorder

### 3.1. Food Selectivity (FS) in Adults with Autism Spectrum Disorder

FS, also termed food fussiness, selective eating, and picky/fussy eating [72], can be defined as when an individual consumes “an inadequate variety of foods.” It is characterized by the consumption of a limited food repertoire associated with aversions to specific temperatures, textures, flavors, colors, and odors [73,74]. Bandini et al. [75] introduced a classification of FS into three domains: food refusal, limited food repertoire, and high frequency of food intake. Food refusal is defined as the number of food items an individual refuses to eat or as the percentage of food items refused among those offered. Limited food repertoire was defined as the number of unique food items consumed over a 3-d period. High-frequency single food intake was associated with a single food item eaten 4–5 more times daily.

Numerous studies have focused on FS in children with ASD [76,77,78,79], demonstrating that FS is more common in this population than in typically developing children [76,77], with rates as high as 85% [78]. This limited food repertoire is linked to nutrient inadequacies [76,77], and research suggests that FS may persist into adulthood if left untreated [36,80,81]. Despite the apparent significance of this issue across the lifespan, FS in adults with ASD has only been explored in six relevant studies, each highlighting different aspects of eating behaviors, sensory sensitivities, and social impacts. One of those studies is based on an intervention design [36], another is a quantitative study using self-report questionnaires [33], and the remaining ones are qualitative studies based on interviews [29,31,34,35], thus offering a diverse perspective on this topic (Appendix A).

Several studies emphasize the role of sensory sensitivities in FS among autistic individuals. Kuschner et al. found that individuals with ASD exhibited higher food neophobia and a greater aversion to textured foods and strong tastes compared to typically developing peers [33]. Similarly, Kinnaird et al. reported that autistic adults avoided certain foods due to hypersensitivity to taste, texture, smell, and temperature, often preferring to eat the same foods repeatedly [29]. Waldron et al. reinforced those findings, noting that many participants restricted their diets due to sensory issues, particularly avoiding soft or stringy foods [31]. These studies collectively suggest that sensory processing differences are a primary driver of FS in autistic individuals.

A key area of comparison is how dietary patterns evolve. Barbier found that while autistic adults initially followed a diet high in carbohydrates and fats with limited vegetable intake, their variety of food choices improved with age, suggesting a certain degree of adaptation [34]. Folta et al. supported this notion, as their study participants reported that selective eating became less restrictive with age; moreover, they developed coping strategies to manage food preferences [35]. In contrast, Kinnaird et al. found that while some participants became more flexible over time, others maintained strong preferences and food avoidance patterns, demonstrating variability in the ways autistic individuals adapt to dietary changes [29].

FS can influence social interactions, yet the extent of its impact varies across studies. Kuschner et al. linked food neophobia to lower adaptive daily living skills, implying broader functional challenges [33]. In contrast, Folta et al. found that selective eating did not significantly impact social participation, as participants developed effective coping strategies [35]. Similarly, Kinnaird et al. reported that although autism influenced eating behaviors, most participants did not perceive their selectivity as a major problem in social settings [29]. These findings suggest that while FS may affect daily life, many autistic adults find ways to manage it without severe social consequences.

While most studies focus on self-reported behaviors, Pubylski-Yanofchick et al. [36] examined an intervention approach, demonstrating that behavioral treatments such as differential reinforcement (positive and negative) effectively increased food acceptance in an autistic adult. The study found that positive reinforcement was preferred and that the treatment generalized to novel foods and settings [36]. This contrasts with other studies, where participants often did not express interest in changing their eating habits. For example, Folta et al. found that participants were not interested in receiving help for their selective eating, indicating a difference in motivation or perceived need for intervention [35].

Across studies, several obstacles to maintaining a balanced diet emerged. Waldron et al. highlighted that limited cooking skills posed a significant challenge, preventing participants from preparing healthier meals [31]. Barbier noted that family encouragement influenced food choices, suggesting that external support plays a role in dietary habits [34]. Meanwhile, studies like Kinnaird et al. and Kuschner et al. pointed to cognitive rigidity as a factor that may contribute to resistance toward dietary changes [29,33].

### 3.2. Food Sensory Processing in Adults with Autism Spectrum Disorder

Although FS is a common challenge in autistic adults, its impact and management vary widely. Sensory sensitivities appear to be a major contributing factor. FS may be associated with differences in sensory perception with an increased or diminished sensitivity to environmental sensory stimuli, as has been investigated in children. Eating is a complex behavior that involves perceptual, emotional, cognitive, and neurological processes. On the afferent perceptual level, the processes involved are distal (visual, auditory, ortho-nasal olfaction) and proximal (gustatory, retronasal olfaction, tactile).

Atypical sensory experiences are reported to occur in as many as 90% of individuals with ASD. They may affect almost every sensory modality: vision, audition, smell, touch, and taste. Hyper or hypo-reactivity to sensory input, as well as unusual interests in specific types of environmental stimulation (e.g., apparent indifference or adverse responses to pain/temperature/sounds and/or textures, atypical smelling or touching of objects, lights or movement, and so on), are also characteristic features of ASD [82]. Research results on sensory processing in autistic population, mainly children, show inconsistent findings across odor detection and identification [33,83,84,85,86,87,88,89,90,91], visual processing of food stimuli [92], and textural perception of food in the mouth [33,93,94,95,96], but overall suggests that atypical sensory responses are closely linked to food refusal behaviors.

Studies on the relationship between food sensory perception and eating behaviors specifically in adults with ASD are scarce and have only begun to appear recently; food sensory perception has become a topic of interest. Our review identified eight relevant studies addressing this emerging area of interest (Appendix A). Tavassoli and Baron-Cohen [37] investigated taste identification abilities in autistic adults using taste strips. Their findings revealed significantly lower taste identification accuracy in ASD individuals compared to controls, particularly for bitter, sour, and sweet tastes. Error analysis suggested that autistic participants frequently misidentified tastes as salty or tasteless. In contrast, Avery et al. [40] utilized fMRI imaging to explore neural correlates of taste perception, finding no significant differences in hemodynamic responses to gustatory stimuli between ASD and control groups. However, increased insular cortex activation in ASD individuals was linked to heightened taste reactivity, suggesting that while behavioral identification of tastes may be impaired, underlying neural processing differences may contribute to atypical eating patterns.

Tavassoli and Baron-Cohen [38] examined olfactory detection thresholds and adaptation in ASD adults. Contrary to expectations, no significant differences were found between autistic and neurotypical (NT) individuals in either threshold detection or adaptation to olfactory stimuli. This finding contrasts with previous reports suggesting olfactory hypersensitivity in ASD, implying that sensory differences in autism may be more domain-specific rather than a universal impairment.

Several studies have linked sensory sensitivity to eating disorder behaviors in ASD populations. Nisticò et al. [43] found that hypersensitivity in vision and hyposensitivity in taste were significant predictors of eating disorder (ED) symptoms in autistic adults. Similarly, Nisticò et al. [44] observed strong associations between hypersensitivity in touch and vision and eating disturbances in young adults. These findings suggest that altered sensory perception contributes to restrictive eating behaviors, potentially increasing the risk of eating disorders in ASD individuals.

Brede et al. [41] provided further qualitative insight, highlighting that food-related sensory sensitivities, such as texture, taste, and temperature aversions, were persistent contributors to restrictive eating behaviors in autistic women. Their study emphasized that these sensitivities often predate the onset of Anorexia Nervosa (AN) and persist even after recovery, reinforcing the role of sensory processing in the maintenance of ED behaviors.

Mayer [39] examined sensory processing patterns across both NT and ASD populations, finding that sensory atypicalities were not exclusive to ASD individuals but followed a gradient related to autistic traits. Notably, lower sensation-seeking behaviors were observed in individuals with increased autistic symptomatology, further supporting the link between sensory processing and selective eating behaviors. Singh and Seo [42] expanded on this by investigating firsthand accounts of sensory-related eating experiences, revealing that hypersensitivity to environmental factors (e.g., noise, lighting, and utensil conditions) significantly influenced food choices and mealtime experiences.

Across studies, sensory sensitivity consistently emerges as a key factor in the development and maintenance of atypical eating behaviors in ASD. While some studies provide evidence for altered taste processing in ASD, the extent of these differences remains debated [37,40]. The role of sensory sensitivity in EDs, outlined by other studies [41,43,44], highlights a potential clinical target for intervention. Additionally, research suggests that sensory differences extend beyond autism, influencing eating behaviors across the broader population [39,42]. Future investigations should aim to clarify the mechanisms underlying sensory-related eating challenges and develop tailored interventions designed to support autistic adults experiencing restrictive eating patterns.

Regarding sensory aspects, very few papers have investigated food sensory aspects in adults, and those that exist were performed with considerable methodological differences on a low number of individuals or performed as preliminary studies. As a consequence, results are not conclusive. These preliminary studies suggest that adults with ASD exhibit dislikes for foods with particular textures and strong tastes. Moreover, one can note that these findings are not fully representative of the ASD spectrum; they mainly involve people with high-functioning autism.

Food sensory sensitivities and correlated food selectivity can lead to an inadequate nutritional status in individuals with autism, potentially resulting in deficiencies or imbalances that may affect overall health and well-being. Understanding the long-term implications of food selectivity is crucial, particularly in autistic adults, as research in this population remains limited. While childhood studies have extensively documented dietary patterns and nutritional concerns, less is known about how these issues persist or evolve in adulthood. Given that nutrition plays a fundamental role in physical and mental health, it is essential to examine how food selectivity influences the dietary intake and nutritional status of autistic adults. Exploring potential deficiencies, dietary imbalances, and their consequences will provide a clearer understanding of the specific nutritional challenges this population faces.

## 4. Nutritional Status in Adults with Autism Spectrum Disorder

Adequate nutritional status implies a balance between nutrient intake and specific nutritional requirements, and should allow for the utilization of nutrients to maintain reserves and compensate for losses. Consequentially, malnutrition may be due to consuming too little or too much of a nutrient or type of food, or it may come from poor food, diet, or supplement choices. However, it is well-known that nutritional status is affected by numerous factors. Some individuals become over- or undernourished due to physiologic or metabolic differences that affect nutrient bioavailability, altered nutrient or energy needs, mental and physical health problems, use of medications, etc. Given the antecedents in children with ASD, health professionals consider autistic adults to be nutritionally vulnerable or at risk for malnutrition. However, as far as we know, there is no review of scientific evidence of the problem’s extent or an assessment of individual risk in this population group.

The concept of evaluating nutritional status implies considering the diversity of factors and the variety of mechanisms involved in achieving nutritional balance. Therefore, nutritional status assessment includes evaluation across at least four different domains, popularly known as anthropometric, biochemical, clinical, and dietary; that is why this procedure used to be named the A-B-C-D assessment [97]. Anthropometry consists in measuring body size, composition, weight, and proportions; on the other hand, biochemical assessment involves measuring nutritional markers and indicators of organ function in biological specimens (blood, urine, feces, hair, nail, and tissue samples). The nutrition-focused clinical (also known as physical) exam assesses a patient for signs and symptoms indicating malnutrition or specific nutrient deficiencies. Dietary intake analysis identifies a patient’s habitual pattern of intake, food preferences (including ethnic, cultural, and religious influences), and dietary habits, but also a quantitative evaluation of nutrients and energy intake, with the aim of estimating the patient’s diet’s adequacy regarding specific nutritional recommendations. Some professionals add information on environmental factors such as socioeconomic status, social support systems, lifestyle, and social interactions that affect nutritional status, regarding them as integral to this nutritional assessment. We have organized this review according to the four primary measurement methods, although, when necessary, environmental factors will be mentioned.

### 4.1. Anthropometric Measurements of Adults with Autism Spectrum Disorder

Table 2 summarizes data regarding weight status in autistic adults from 13 studies published between 2008 and 2024, predominantly conducted in the United States [23,29,46,47,48,52,53,54,55], with additional data from Canada [45], Sweden [49], Spain [50], and the UK [29]. Sample sizes varied considerably, from 34 to 6019 autistic adults. The mean age across studies ranged from 24 to 41 years, with one study focusing specifically on older adults (≥65 years) [29]. Male representation was predominant in most studies, except for the study of Sedgewick et al., which had only 15.8% males [51]. The prevalence of ID varied substantially across studies (0–100%). Unfortunately, none of these studies reported their participants’ level of ASD.

Obesity prevalence ranged from 10% to 51.4%, with most studies reporting rates between 27 and 37%. Overweight rates typically ranged from 18 to 29%, with one Spanish study reporting 47.1% [50]. Studies incorporating control groups consistently demonstrated higher obesity rates in autistic adults [7,13,49,51]. Based on one study’s results, the prevalence of obesity was apparently different according to age group [48]. Regarding differences by sex, several studies found higher obesity rates among autistic females compared to males [7,13,49]. However, others found similar figures [52,54] or an even higher combined overweight/obesity prevalence in men [50]. Interestingly, one study observed an inverse relationship between obesity prevalence and ID severity, with higher rates in autistic subjects without ID compared to those with ID [55]. The reporting of underweight status (BMI ≤ 18.5 kg/m^2^) was less consistent across studies, with only five studies including this category. The reported prevalence of underweight ranged from 4.6% to 18% [46,47,51,52,53]. Studies with control groups consistently showed higher underweight rates in autistic adults compared to non-autistic individuals [51,53]. Notably, Weir et al. found that autistic adults were twice as likely to be underweight compared to controls, even after adjusting for potential confounders, including sociodemographic factors, lifestyle habits, and dietary characteristics [53]. Finally, the Spanish study was the only study that included body composition analysis, revealing significant gender differences in body fat percentage [50].

These findings suggest a significant burden of overweight and obesity among autistic adults, with possible age and gender-specific patterns as well as interactions with ID status. These results are in line with the conclusions of a recent meta-analysis that referred to a “worrisome epidemic of obesity and overweight in individuals with ASD” [11]. Considering that obese individuals with ASD show an increased risk of in-hospital mortality, type II diabetes, cardiovascular disease, and other co-occurring health conditions [98], strategies to identify excessive weight are an urgent task. In parallel, a variable but relatively high prevalence of underweight was observed. We thus conclude that weight-management challenges in autistic adults span the entire weight spectrum, not just overweight and obesity. In addition, the higher prevalence of both underweight and overweight/obesity compared to control populations points to a more complex relationship between autism and weight regulation that warrants particular attention in clinical practice and public health interventions [99]. Consequently, future research should focus on how to manage the specific factors contributing to both extremes of the weight spectrum in this population.

In any case, it is difficult to establish general conclusions about weight status in autistic adults due to several methodological limitations. Data collection methods varied across studies, including medical records, questionnaires, and direct measurements, and this may have contributed to the variability we observed in reported prevalence rates. Sample sizes and representativeness also differed substantially among studies: Geographical variations might reflect broader population-level differences in inadequate weight prevalence and different sociocultural factors affecting weight status; The substantial variation in gender distribution across studies may partly explain the differences in prevalence rates; The proportion of participants with ID appears to influence obesity rates. These patterns suggest that the variability in reported weight status prevalence in these studies likely reflects actual population differences along with methodological factors, highlighting the need for standardized assessment approaches in future research.

However, it must be emphasized that these results are based solely on BMI, which implies a limited understanding of the actual situation. While BMI is widely used for population-level assessment, it does not distinguish between fat mass and lean mass, potentially misclassifying individuals, particularly in populations with altered body composition patterns and/or weight-related morbidity. Recently, the Lancet Diabetes & Endocrinology Commission introduced a crucial paradigm shift in obesity diagnosis and classification, highlighting the limitations of using BMI as the sole diagnostic criterion [100]. The new framework distinguishes between Clinical Obesity, characterized by organ dysfunction due to excess adiposity, and Preclinical Obesity, marked by preserved organ function but increased disease risk. This distinction requires a comprehensive assessment approach incorporating anthropometric and clinical components that go beyond mere BMI measurements. For autistic adults, where gender-specific differences in body composition have been documented despite similar BMI classifications, this comprehensive approach becomes especially critical.

### 4.2. Biochemical and Clinical Assessment of Nutritional Status in Adults with Autism Spectrum Disorder

Biochemical assessment is enormously important, as it allows researchers and clinicians to evaluate nutritional deficiencies while detecting subclinical malnutrition states that could be highly prevalent in this population. However, we only found one study [56] in which autistic adults’ nutritional status was assessed using biomarkers (Appendix A). That population-based cross-sectional study conducted in the Faroe Islands examined 25-hydroxyvitamin D3 (25(OH)D3) levels in a complete cohort of 40 young adults (15–24 years) with ASD, comparing them with their typically developing siblings and parents, and age- and gender-matched healthy controls. Results revealed significantly lower 25(OH)D3 levels in the ASD group compared to all the other groups, with 88% of individuals with ASD showing vitamin deficiency. This is the first study examining vitamin D levels in a complete population of young adults with ASD, complementing previous literature focusing on pediatric populations [12,101]

We only found one study that evaluated the nutritional status of autistic adults from a clinical point of view. Croen et al. [7] conducted a cross-sectional, case-control study in the United States to assess the health status of adults on the AS (Appendix A). The study included 1507 adults with formally diagnosed ASD, of whom 27% were female. The sample was ethnically diverse and encompassed adults across all ages and varying levels of intellectual functioning. Findings revealed that adults with ASD had a significantly higher prevalence of vitamin deficiency compared to a control group of 15,070 age- and sex-matched typical adults, with autistic females showing a notably higher deficiency rate than males (9.6% vs. 3.3%). It was not reported which vitamins were considered. As the authors pointed out, the overall prevalence of vitamin deficiency was low relative to other medical conditions; however, a certain degree of underestimation was plausible due to the dependence on the content incorporated in clinical databases versus direct assessments from participants. Additionally, a lack of information about the use of supplementation did not allow those researchers to establish the actual existence of vitamin deficiencies.

These biochemical and clinical assessment results indicate significant knowledge gaps regarding potential nutritional deficiencies in adults. Current findings suggest that many aspects of adult nutritional status remain poorly understood, necessitating further comprehensive research to establish a complete picture in this regard.

### 4.3. Dietary Assessment of Nutritional Status in Adults with Autism Spectrum Disorder

Among the studies we reviewed, six articles included information about dietary habits and/or dietary quality in autistic adults [53,57,58,59,60,61] (Appendix A).

Two studies describing educational interventions for American autistic adults included information related to diet as part of the sample description. The first of them revealed that more than a third of participants, 13 young adults with level 1 ASD, acknowledged that their current diet was unhealthy. All participants reported consuming fast food or take-out, with 54% reporting that they ate fast food or take-out several times per week [61]. The second article, studying the dietary patterns among seven university students with ASD, found relatively high consumption of fruits (5–6 times per week) and moderate vegetable intake (3–4 times per week in mixed dishes), with notably low consumption of cooked dried beans. French fries and potato chips were among the most frequently consumed items [60]. These studies share methodological limitations that impact their generalizability. Both studied small sample sizes with very specific characteristics and relied exclusively on self-reported data collected through non-validated questionnaires, without employing complementary data collection methods or external validation processes to verify the accuracy of participants’ responses.

A cross-sectional case-control study in Sweden with 47 adults formally diagnosed with ASD and 69 matched controls evaluated dietary habits predictive of dental caries. The ASD group exhibited less frequent snacking than controls (51% vs. 71%). However, both groups showed similar patterns in consuming potentially cariogenic foods, including candies, pastries, chips, and sweetened drinks, with no significant differences between the two groups [57]. At any rate, certain limitations, including self-reported data and a small sample size (the ASD group consisted of only those who accepted dental examination invitations, representing less than 12% of contacted individuals), significantly impact the generalizability of these findings.

Kranz et al. conducted a cross-sectional population-based study aimed at examining dietary patterns in a sample of 488 American young adults with ASD formally diagnosed (aged 18 to 28) through their parents’ reports [59]. Most of those parents’ adult children (76.8%) consumed a varied diet and avoided eating the same foods daily (66.8%). Snacking between meals was common (75.8%), with a preference for sweet (44.4%) and salty (26.2%) snacks over healthier alternatives like fruits and vegetables (7.4%). Regarding hydration, parents reported that only half of their adult children met the recommended daily water intake, with a clear preference for cold beverages (39.8%) over hot drinks (12.9%). Parents also noted that nutritional supplementation was common, with half of their adult children using supplements, particularly multivitamin and mineral combinations (38.9%). A notable finding regarding dietary development showed that while parents identified 52% of their children as picky eaters in childhood, they reported that nearly half (48.2%) now consumed previously rejected foods. According to parental observations, current food group preferences demonstrated specific patterns: their children’s vegetable consumption focused on white/starchy (22.1%) and orange varieties (21.7%), protein intake was dominated by red meat (29.4%) and poultry (27.2%), grain consumption was balanced between white (24.6%) and whole grain (21.8%) options, and fat intake primarily came from vegetable oils (31.1%) and butter (24.8%). Unfortunately, information about other essential food groups, such as fruits, fish, and dairy products, is lacking in this study. Anyway, these figures must be considered with caution because, as mentioned, they relied on parental self-reported data and were therefore potentially biased. They were obtained by a non-validated questionnaire, and the authors did not use dietary collection data or other means to validate the information provided by the parents.

Finally, we found two large-scale cross-sectional studies specifically designed to describe the diet quality of autistic adults. A transnational study by Weir et al. [53] found that autistic adults (*n* = 1183 ASD formally diagnosed) were more likely to have dietary restrictions and preferences, including vegan, vegetarian, lactose-free, and gluten-free diets. It also showed a higher tendency to frequently consume high-calorie, high-fat, salty, or sugary foods. Autistic females, but not males, specifically demonstrated lower consumption of fruits and vegetables compared to sex-matched peers. Regarding hydration, no differences were found regarding the likelihood of meeting daily water goals (i.e., >8 cups/glasses) or drinking high-sugar beverages (soft drinks, fruit juice, smoothies, etc.) frequently, but autistic adults reported drinking less caffeinated beverages (tea, coffee, energy drinks, etc.) [53]. Again, despite being the most extensive study of its kind, conclusions may not be generalized due to several limitations regarding sample representation. The sample was biased toward females (63%), predominantly white participants with higher education levels, and UK residents. Besides, the recruitment process through social media and autism support groups, combined with the requirement for internet access, may have excluded certain demographic groups, particularly those with moderate to severe ID [102]. From the point of view of nutritional status assessment, another limitation is that the authors unfortunately did not provide descriptive data about these variables to establish the adequacy of autistic diet.

As a sub-study of the Eating Habits and Well-Being study on Japanese manufacturing workers, Nakamura et al. [58] investigated 2053 adults with autistic traits, finding gender-specific dietary patterns. Men with higher autistic traits displayed a lower intake of iron, vitamin B12, seaweed, and fish/shellfish. Women with higher autistic traits demonstrated increased carbohydrate intake but lower consumption of proteins, fats, minerals, vitamins, and dietary fiber [58]. The study’s primary limitation is the absence of absolute food consumption frequency data, presenting only sex-based comparisons. This limitation makes it impossible to assess whether the diets were nutritionally adequate despite the observed differences between groups. Furthermore, the conclusions must be considered with caution as the study population consisted of individuals with autistic traits rather than diagnosed autism. While these findings can serve as a proxy result, they primarily highlight the pressing need for comprehensive nutritional intake assessment in adults with ASD.

To sum up, multiple studies indicate suboptimal dietary patterns, particularly high-calorie, nutrient-poor food choices. Fast food consumption and processed foods appear to be consistent concerns across studies. Lower intake of fruits and vegetables emerges as a common pattern. Studies that analyzed gender differences consistently found slightly distinct dietary patterns between males and females. Based on our comprehensive review, we cannot establish whether nutrient intake inadequacies are present in formally diagnosed adults with ASD, as none of the reviewed studies provide information in this regard.

## 5. Dietary and Nutritional Intervention Studies

As mentioned in previous sections, individuals with ASD often exhibit strong food preferences, food sensory sensitivities, and rigid eating behaviors, which can lead to unbalanced diets, nutritional deficiencies, and an increased risk of metabolic disorders, such as obesity and diabetes. Consequently, dietary interventions explicitly designed for this population have become an area of interest in the scientific community. Such dietary interventions generally follow two approaches: therapeutic dietary protocols and educational interventions designed to promote healthy eating patterns. Notable therapeutic approaches—including gluten-free/casein-free diets, specific carbohydrate protocols, intermittent fasting regimens, and ketogenic dietary patterns—have demonstrated varying degrees of efficacy in mitigating gastrointestinal symptomatology and behavioral manifestations [22,26,103,104,105,106,107,108,109,110]. Nutritional supplementation strategies, particularly those involving omega-3 fatty acids, vitamin/mineral complexes, and pro/prebiotics interventions, have yielded modest improvements in core ASD symptoms, cognitive function, and gut–brain axis regulation [22,26,111,112,113]. Additionally, educational interventions designed to expand dietary repertoire, enhance adherence to a healthy diet, and ameliorate problematic mealtime behaviors have been implemented with promising preliminary outcomes [72,114]. Nevertheless, a significant research gap exists regarding dietary modification strategies specifically tailored for adult ASD populations, with the current literature predominantly focused on pediatric interventions.

### 5.1. Impact of Nutrient Supplementation or Specific Diets in Adults with Autism Spectrum Disorder

Five studies on adults with ASD specifically evaluate either the effect of dietary nutrient supplementation or the impact of specific diets (Appendix A). Among those five studies, three of them assessed the impact of supplementation or diets through surveys evaluating beneficial and adverse effects [62,64,66], while the two other ones were based on interventions [63,65]. Although these are the only studies that include an adult population with ASD, only one of them, the investigation that explored the effects of *n*-3 long-chain polyunsaturated fatty acids (*n*-3 LCPUFA) on cognitive functions in adults with ASD, was conducted exclusively on adults [63]. The remaining studies included both adult and pediatric populations, with adults and adolescents comprising only 30% of the sample as a maximum percentage; however, only one of them featured a sub-analysis of data based on participants’ age [62]. Therefore, these studies’ results and conclusions [64,65,66] need to be explicitly validated in adult populations. Additionally, the study by Geng et al. does not specify the exact number of subjects with ASD, as its primary focus was on individuals with communication delays and motor function impairments [64].

Despite these sample limitations, results from intervention studies concerning supplementation suggest promising outcomes. Fish oil supplementation may improve attention and working memory in adults with ASD and alleviate ADHD symptoms in those with comorbid ADHD [63]. Similarly, sequential supplementation (vitamin/mineral supplement-day 0, essential fatty acids-day 30, Epsom salt baths-day 60, carnitine-day 90, digestive enzymes-day 180, and a healthy gluten-free/casein-free/soy-free (HGCSF) diet-day 210) could be responsible for significant improvements in nonverbal IQ, ASD symptoms, and nutritional biomarkers [65]. In their survey-based study, Geng et al. [64] collected self-reported data on the effects of a multi-nutrient supplement (IQed) in individuals with speech and motor impairments; the vast majority of respondents reported positive changes in behavior or physical symptoms. Similarly, Adams et al. [62] assessed the ANRC Essentials Plus supplement through a retrospective survey, finding a 73% positive response rate. However, as in Adam’s intervention study [65], response bias, lack of objective validation, and the inability to isolate specific ingredient effects limit the general validity of its conclusions.

A recent survey-based study presented findings on the efficacy of 13 therapeutic diets for ASD based on responses from 818 survey participants in the United States [66]. The study evaluated the benefits, adverse effects, and symptom improvements associated with these diets. Their average overall benefit was substantially higher than that of nutraceuticals and psychiatric/anticonvulsant medications. The study found that different diets affected different symptoms, suggesting that individual symptom profiles could guide the selection of the most effective dietary interventions. While the reported benefits lack objective validation, the findings suggest that therapeutic diets can improve certain ASD-related symptoms with minimal adverse effects.

In addition to the reported limitations, the main constraint of all these studies, in relation to our review, is that none of them assessed the impact of supplementation or the studied diets on the nutritional status of autistic adults. Instead, they focused on evaluating the impact of dietary interventions on autism-related symptoms.

### 5.2. Dietary Intervention to Implement a Healthy Diet for Adults with Autism Spectrum Disorder

Encouraging dietary patterns that prioritize nutrient-rich foods could be a feasible and beneficial approach, supporting overall well-being while minimizing potential nutritional risks associated with food selectivity. ASD associations and nutritionists have agreed that promoting adherence to a healthy diet may be a valuable strategy in autism [115,116,117,118]

Very few studies have explored different approaches to improving the dietary habits of adults with ASD, ranging from modified meal plans in collective catering settings to cooking courses aimed at increasing food autonomy. Among these, five key studies offer valuable insights into how diet can be adapted to meet the nutritional and sensory needs of autistic individuals while also promoting healthier eating habits (Appendix A).

Conti et al. examined a two-phase intervention in an Italian daycare facility for adults with ASD [67]. The first phase consisted of an observational study analyzing the participants’ food intake, meal preferences, and nutritional deficiencies. This was followed by an intervention phase in which modified menus were introduced to better align with the individuals’ sensory preferences and dietary needs. The results showed that autistic adults tended to favor foods with soft textures, mild flavors, and neutral colors. By incorporating these preferences into meal planning, the study found a significant improvement in meal acceptance and a reduction in food waste, highlighting the importance of tailoring diets to the sensory profiles of individuals with ASD. A previous review of this group [115] that included the role of meal planning and food presentation, confirmed that personalized menus that take sensory preferences into account lead to better meal acceptance. The researchers also emphasized the need for structured mealtime environments, as inconsistencies in food presentation and eating routines contribute to meal rejection. Their work suggests that the consistency of meals, both in content and in the environment in which they are consumed, plays a crucial role in dietary interventions for autistic adults.

A particularly innovative study was conducted by Veneruso et al., who introduced Il Tortellante^®^, an Italian culinary project designed to promote adaptive behavior, enhance social skills, and reduce ASD-related symptoms in adolescents and young adults [68]. This longitudinal study focused on a group-based cooking intervention in which participants learned to make fresh pasta by hand, allowing them to develop practical and social skills in a structured environment. The study results indicated that participants showed significant improvements in daily living skills, social engagement, and a reduction in ASD-related symptom severity. The success of this culinary program suggests that hands-on, interactive food preparation activities not only help autistic individuals develop independence in the kitchen but also contribute to broader developmental gains, including enhanced communication and cooperation in group settings.

A third study by Nabors et al. consisted of a pilot study examining the effects of a year-long healthy eating and exercise program tailored to young adults with ASD and ID [69]. The intervention combined nutrition education, exercise training, and motivational interviewing to set individualized health goals for participants. The program included lessons on balanced nutrition using concepts such as MyPlate and the food pyramid, as well as education on vitamins, minerals, and portion sizes. Additionally, participants were introduced to various types of physical activity and learned about the benefits of regular exercise. The study had promising results, as two participants experienced significant weight loss, while others successfully maintained their weight despite the challenges posed by the COVID-19 pandemic. Participants and their parents reported increased knowledge about healthy eating and positive behavior changes, although the authors acknowledged the need for further research with control groups and long-term follow-up to confirm these findings.

In addition to those studies, Gustin et al. focused on improving food autonomy by teaching young adults with ASD how to cook [70]. Conducted as a six-week course, this intervention provided structured cooking lessons that incorporated visual guides and step-by-step instructions to accommodate the learning styles of autistic individuals. Pre- and post-course assessments revealed that participants not only increased the frequency of their home-cooked meals but also gained confidence in their ability to prepare a variety of dishes. This finding is particularly relevant, given that many autistic adults struggle with independence in daily living skills, and the ability to prepare meals can significantly enhance their quality of life.

Another innovative study took a different approach by applying behavioral economics principles to encourage healthier food choices among autistic adolescents and young adults in a residential school setting [71]. Their Smarter Lunchroom intervention used subtle environmental changes, such as making fruits and vegetables more accessible and using visual cues to promote healthy options. The intervention resulted in a notable increase in the consumption of whole grains and fresh produce while also reducing the intake of refined grains. Food waste was significantly reduced, suggesting that simple changes in how food is presented can have a profound impact on eating behaviors among individuals with ASD. These findings reinforced the idea that people with ASD benefit from structured environments that reduce cognitive overload and support predictable decision-making processes. By making small, strategic changes in food presentation, the researchers were able to improve dietary quality without forcing individuals to abandon their food preferences abruptly.

Taken together, these studies illustrate that dietary interventions for adults with ASD are most effective when they incorporate elements of sensory adaptation, structured meal planning, and behavioral strategies that promote gradual change. A key takeaway from this body of research is that restrictive diets or forceful interventions are unlikely to succeed; rather, success comes from understanding and working within the unique sensory and behavioral characteristics of autistic individuals.

The studies we reviewed provide compelling evidence that personalized dietary strategies, cooking interventions, and environmental modifications can substantially enhance the dietary habits of adults with ASD. Future efforts should aim at implementing policies to develop standardized dietary guidelines for this population.

## 6. Conclusions and Research Gaps

Our literature review on food selectivity in autistic adults highlights a clear trend toward selective eating, primarily influenced by sensory perceptions of food. Despite the limited number of available studies, our observations consistently indicate that sensory characteristics such as taste, texture, smell, and temperature play a crucial role in food choices among autistic adults. This selectivity has direct implications for their nutritional status. Preferences for certain foods and aversions to others can lead to an unbalanced diet, affecting the intake of essential nutrients and overall health.

Despite a general lack of studies on this subject, certain trends are clear: observed overweight and obesity prevalence ranging from 10% to 51.4%, with most studies reporting rates between 27 and 37%. It also appears clear that individuals with ASD have associated nutritional deficiencies. Furthermore, multiple studies indicate suboptimal dietary patterns featuring high-calorie, nutrient-poor food choices. Fast food consumption and processed foods appear to be consistent concerns across studies. A lower intake of fruits and vegetables emerges as a common pattern.

Despite a clear need for improved nutrition among autistic adults, valid dietary-intervention studies are lacking. The few studies related to supplementation in adults are not valid for the purpose of this review, as they do not evaluate the effects of supplementation on the individual’s nutritional status, but study the effects on symptomatology. Research exploring the effects of fatty acid supplementation on cognitive and behavioral outcomes in adults with ASD holds significant theoretical and clinical relevance.

According to the dietary-intervention studies featured in this review, the application of restrictive dietary pattern modifications does not seem to present any conclusive results in the nutrition of autistic adults beyond the clear benefits of improving adherence to a healthy and balanced diet. Encouraging dietary patterns that prioritize nutrient-rich foods could be a feasible and beneficial approach, supporting overall well-being while minimizing potential nutritional risks associated with food selectivity.

Promoting adherence to a healthy diet may be a valuable strategy in autism, and several studies featured in this review indicate that dietary interventions for adults with ASD are most effective when they incorporate elements of sensory adaptation, structured meal planning, and behavioral strategies that promote gradual change. A key takeaway from this body of research is that restrictive diets or forceful interventions are unlikely to succeed; rather, success comes from understanding and working within the unique sensory and behavioral characteristics of autistic individuals.

To improve the nutrition of autistic adults, it is essential to conduct more studies with a methodological approach that reaches clear conclusions. Firstly, nutritional intervention studies should be designed with proper identification of the sample group (sex, age, degree of autism) while including a neurotypical control population and ensuring a sufficient number of participants.

Future studies should incorporate multiple anthropometric measurements (waist circumference, waist-to-hip ratio, waist-to-height ratio) and assess obesity-related organ dysfunction to better characterize the health implications of altered body composition in this population. This approach would align with the new obesity framework while addressing the specific needs of autistic adults [100]. Detecting clinical and subclinical deficiencies in vitamins and minerals through biochemical markers would allow for the implementation of dietary and/or pharmacological measures to correct these deficiencies, benefiting both general health and this group’s specific circumstances. The clinical evaluation of autistic adults should always include a systematic assessment of their most prevalent nutritional disorders.

Moreover, it is crucial to address the scarcity of information on the diet of autistic adults. Studies should emphasize the need for continued dietary support and intervention throughout the lifespan, not just in childhood. Dietary interventions may need to be gender-specific and should consider nutritional adequacy as well as individual dietary preferences/restrictions. Researchers should be cautious of significant methodological limitations, such as reliance on self-reported data without external validation and the use of non-validated tools due to the lack of standardized dietary assessment methods specific to autistic adults [119,120]. It is also important to ensure sample representativeness. Larger, more representative studies with diverse samples using validated assessment tools specific to autistic adults are needed to evaluate all relevant food groups and consider factors affecting dietary choices. This will help establish the nutritional adequacy of diet habits of adults with ASD and develop a comprehensive understanding of their actual nutritional status.

Finally, dietary interventions tailored to sensory preferences and nutritional needs should be implemented to improve food acceptance and reduce waste in adults with ASD. Cooking education programs can enhance dietary autonomy and confidence, fostering independence in food preparation. Behavioral economics strategies, such as food placement and visual prompts, should be used to encourage healthier eating habits without restricting choices. Environmental modifications in meal settings, including structured food presentations and meal planning, can help reduce food neophobia. Future research should expand sample sizes, employ longitudinal studies, and refine interventions to ensure broader applicability for autistic adults.

## Figures and Tables

**Table 1 nutrients-17-01456-t001:** Summary of 43 reviewed studies focusing on food and nutrition in autistic adults.

Reference	Aim	Country—Study Design and Sample
Food selectivity in autistic adults		
Kuschner et al., 2015 [33]	To examine self-reported food selectivity in adolescents and young adults with ASD and compare it to typically developing controls.	USA—Preliminary quantitative study. 65 adolescents/young adults with ASD: 11% females, aged 12 to 28, broad ASD, IQ ≥ 75 59 typically developing controls matched on age, IQ, and sex ratio.
Barbier, 2015 [34]	To qualitatively explore the relationship between eating behaviors and autism using the SWEAA questionnaire and interviews with adults diagnosed with ASD.	USA—Qualitative study. 4 male participants aged 22–27.
Kinnaird et al., 2019 [29]	To explore whether autism impacts eating for some autistic individuals in adulthood and to what extent those individuals perceive this as a problem.	UK—Exploratory qualitative study. 12 adults with ASD: 68% females, aged 38.5 + 13.9 (range 19–71) years.
Folta et al., 2020 [35]	To explore the impact of selective eating on key social interactions—with family, peers, and in other social situations—of transition-age autistic youth who self-identified as food-selective.	USA—Qualitative study. 20 autistic young adults: 20% females, aged 18–23.
Pubylski-Yanofchick et al., 2022 [36]	To evaluate behavioral treatments to increase the acceptance of novel foods, specifically fruits and vegetables, in an adult with ASD.	USA—Intervention study: Combined alternating-treatments and changing-criterion design. A 26-year-old male with ASD who had a long history of food selectivity.
Waldron et al., 2022 [31]	To describe self-care practices of adults with ASD and explore the self-reported impact of such practices on their health and well-being.	USA—Exploratory qualitative study. 29 adults with ASD, 77% diagnosed by a medical professional: 43% females, aged 51 to 79.
Sensory processing of food in autistic adults		
Tavassoli & Baron-Cohen, 2012 [37]	To investigate taste identification accuracy and error types in adults with ASD using “Taste Strips.”	UK—Comparative study. 23 adults with ASD (21 with Asperger Syndrome, 2 with High-Functioning Autism), mean age 35.8. 26 control participants with no psychiatric condition, aged 25.1 years.
Tavassoli & Baron-Cohen, 2012 [38]	To investigate olfactory detection thresholds and adaptation to olfactory stimuli in adults with ASD.	UK—Comparative exploratory study. 38 adults with ASD aged 35.9 years and 42 control participants aged 28.8 years. A subgroup of 19 participants from each group conducted an adaptation task.
Mayer, 2017 [39]	To examine specific patterns across autistic traits and sensory behaviors within both ASD and neurotypical populations.	UK—Correlational study. Recruited 580 NT adults and 42 high-functioning ASD adults with a confirmed diagnosis. Mean age of 35.07 (SD = 12.38).
Avery et al., 2018 [40]	To examine the neural correlates of taste reactivity in individuals with ASD and compare them with typically developing (TD) controls.	USA—Comparative exploratory study. 21 males with ASD and 21 TD males, aged 15 to 29 years., with a mean age of 21 years Participants with ASD met DSM-5 criteria and had IQ scores ≥ 80.
Brede et al., 2020 [41]	To better understand how Anorexia Nervosa develops and persists in autistic individuals from the perspective of autistic women, parents, and healthcare professionals.	UK—Qualitative research. 15 autistic women, mean age 32.6 13 parents of autistic women and 16 healthcare professionals.
Singh & Seo, 2022 [42]	To achieve a better understanding of how atypical eating behaviors might be associated with specific sensory functions and consumption environments through firsthand accounts of autistic individuals.	USA—Exploratory study. 23 autistic adults aged 19 to 55, with a mean age of 26
Nisticò et al., 2023 [43]	To evaluate the relationship between sensory sensitivity and autistic eating behaviors or eating disorders symptomatology in adults with ASD without ID.	Italy—Observational exploratory study. 75 adults with ASD without ID, mean age 36.1
Nisticò et al., 2024 [44]	To investigate the prevalence of eating disorder symptomatology and its potential relationship with autistic traits and sensory sensitivity in young adults.	Italy—Observational exploratory study. 259 young adults aged 18 to 24
Anthropometric measurements of autistic adults		
Eaves & Ho, 2008 [45]	To learn about the health, physical activity, educational achievement, social adjustment, and quality of life of young adults with ASD.	Canada—Telephone interview with quantifiable questions 48 adults with ASD aged 19 to 31 with a mean age of 24. Males: 77% ID: 83%
Hsieh et al., 2014 [46]	To examine the prevalence of obesity in adults with intellectual disabilities compared to the general population and identify associated factors.	USA—Longitudinal Health and Intellectual Disabilities study. Mail and online surveys 158 adults with ASD aged >18 years. Males: 73% ID: 100%
Croen et al., 2015 [7]	To describe the frequency of medical and psychiatric conditions among a large, diverse, insured population of adults with autism in the US.	USA—Electronic medical records of the Kaiser Permanente health system in Northern California 1507 adults with ASD formally diagnosed: 27% female, aged 29.0 + 12.2 (from 18 to >65) years old, multiracial (White 70%; Black 8%; Asian 11%; Other 11%), at least 19.2% ID 15,070 age- and sex-matched typical controls.
Jones et al., 2016 [47]	To describe medical conditions experienced by a population-based cohort of adults with ASD.	USA—Measurement by study personnel “UCLA-University of Utah epidemiologic survey of autism” 92 adults with ASD. Age (years): Mean: 36; Range: 24–51. Males: 75%. ID: 62.0%
Fortuna et al., 2016 [48]	To examine health conditions and functional status in adults with ASD and identify factors associated with health and functional status across age cohorts.	USA— Medical records from Rochester, NY255 adults with ASD. Age (years): Mean 33.6, 71.8% between 18 and 39. Males 75.3%. ID:50.2%
Flygare Wallén et al., 2018 [49]	To study the prevalence of diabetes and hypertension in persons with ID or ASD.	Sweden—Administrative data for all healthcare consultations from Stockholm County 6019 adults with ASD/1568, 249 Control. Age (years): 89% aged 19 to 49. Males: 60.6% ID: None
Garcia-Pastor et al., 2019 [50]	To compare body composition and physical activity levels between children and adults with ASD.	Spain—Anthropometric measures were assessed for each participant. Special schools and centers for children and adults with ASD 34 adults with ASD. Age (years): Mean: 31.1; SD: 6.9. Males: 67.6% ID: None
Sedgewick et al., 2020 [51]	To examine the relationship between autism and weight outcome in adults.	UK—Questionnaire 335 adults with ASD/330 Control. Age (years): Mean: 34.1; SD: 10.9; Range: 18–71. Males: 15.8%.
Ptomey et al., 2020 [52]	To analyze weight status and associated comorbidities in children and adults with intellectual and developmental disabilities, including Down syndrome and ASD.	USA—University of Kansas Medical Center HERON clinical integrated data repository. 585 adults with ASD aged >18 years. Males: 71.6%
Hand et al., 2020 [13]	To compare the prevalence of physical and mental health conditions in a national sample of autistic older adults to a matched population comparison cohort.	USA—Medicare Standard Analytic Files for the years 2016 to 2017 4685 adults with ASD/46,850 Control. Age (years): 77.5% aged 65 to 74 years; Range: 65 to >84 Males: 67.8% ASD ID: 43.8%
Weir et al., 2021 [53]	To examine whether obesity-related dietary, exercise, and sleep patterns are noted among autistic adults, as well as whether these lifestyle factors contribute to the elevated risks of chronic diseases seen among autistic adults.	Several countries, mainly UK (71%) and USA (10%)—Cross-sectional, case-control study. 1183 participants with ASD formally diagnosed: 63% female, aged 41.0 + 14.4 years old (range 16 to 90)**,** multiracial (White 88%; Multiracial 6%; Other 6%), moderate to severe ID excluded—1.8% ID self-identified, 59% university studies, 18% secondary/high school, and 23% lower level of studies 1203 age-matched typical controls
Thom et al., 2022 [54]	To assess the prevalence of overweight, obesity, and hypertension in a large clinical sample of adults with a confirmed diagnosis of ASD and to examine potential clinical predictors.	USA—Electronic health records of the Massachusetts General Hospital Lurie Center for Autism 622 adults with ASD. Age (years): Mean: 28.1; SD: 7.1; Range: 20–65. Males: 78% ID: 59%
Shameem et al., 2024 [55]	To explore the relationship between intellectual/developmental disability and overweight/obesity and comorbid diagnoses of ASD.	USA—Electronic health record of patients of the Ohio Telepsychiatry Project/Access Ohio clinic 148 adults with ASD. Age (years): >18 Males: NA. ID: 68.2%
Biochemical assessment of autistic adults’ nutritional status		
Kočovská et al., 2014 [56]	To explore vitamin D (25(OH)D3) levels in a population-based study of young adults with ASD in the Faroe Islands and compare them to siblings, parents, and healthy controls.	Faroe Islands (Denmark)—Cross-sectional population-based study. First ever entire population study of ASD individuals’ vitamin D levels (*n* = 219): ⚬ 40 individuals with ASD formally diagnosed: 23% females, aged 15–24 years, % ID not reported ⚬ 62 typically developing siblings: 53% females ⚬ 77 typically developing parents: 52% females ⚬ 40 healthy age- and gender-matched controls
Clinical assessment of autistic adults’ nutritional status		
Croen et al., 2015 [7]	To describe the frequency of medical and psychiatric conditions among a large, diverse, insured population of adults with autism in the US.	USA—Cross-sectional, case-control study. 1507 adults with ASD formally diagnosed: 27% female, aged 29.0 + 12.2 (from 18 to >65) years old, multiracial (White 70%; Black 8%; Asian 11%; Other 11%), at least 19.2% ID 15,070 age- and sex-matched typical controls.
Dietary assessment of autistic adults’ nutritional status		
Blomqvist et al., 2015 [57]	To test if adults with ASD have a higher prevalence of caries, have more risk factors for the development of caries, and apply dental health measures to a lesser extent than people recruited from the normal population.	Sweden—Cross-sectional, case-control study. 47 adults with ASD formally diagnosed: 47% female, aged 33 + 8 years old, without ID, but a wide range of severity (AQ from 5 to 46) 69 age- and sex-matched typical controls.
Nakamura et al., 2019 [58]	To investigate an association between dietary intake and autistic traits.	Japan—Cross-sectional, observational study. Sub-study of the Eating Habit and Well-Being study of Japanese manufacturing workers. 2053 adults, 30% females, with some level of autistic traits assessed using the Japanese version of the Subthreshold Autism Trait Questionnaire, SATQ: ⚬ Male: 20.3% high and 49.5% moderate SATQ score ⚬ Female: 13.5% high and 43.6% moderate SATQ score
Weir et al., 2021 [53]	To examine whether obesity-related dietary, exercise, and sleep patterns are noted among autistic adults, as well as whether these lifestyle factors contribute to the elevated risks of chronic diseases seen among autistic adults.	Several countries, mainly UK (71%) and USA (10%)—Cross-sectional, case-control study. 1183 participants with ASD formally diagnosed: 63% female, aged 41.0 + 14.4 years old (range 16 to 90)**,** multiracial (White 88%; Multiracial 6%; Other 6%), moderate to severe ID excluded—1.8% ID self-identified, 59% university studies, 18% secondary/high school, and 23% lower level of studies 1203 age-matched typical controls
Kranz et al., 2022 [59]	To examine parents’ perception of food intake for themselves and their young adult children with ASD and to explore the potential for perceived intergenerational transfer of dietary intake patterns.	USA—Cross-sectional population-based, explorative pilot study. 488 parents (or primary caregivers) of young adult children with ASD recruited from the “Autism Speaks” database and an internet search of schools/facilities that served them. Young adult children with ASD formally diagnosed by a physician or school psychologist (Level 1: 47.6%; Level 2: 21.7%; Level 3: 11.4%; Missing/not specified: 19.4%): 25.3% female (Missing/not specified: 25.5%), aged 21.8 + 3.7 years old (range 18 to 28), under care in the same residence as the parent (or primary caregiver).
Docherty, 2023 [60]	To develop and implement a curriculum to educate adults with ASD on meal planning and preparation to increase nutrition and cooking knowledge and skills, as well as readiness to change.	USA—Educational intervention study without a control group. 7 university students with ASD enrolled in the Summer LIFE @ the Beach and LIFE Project programs at California State University, Long Beach: 29% female, aged 21.0 + 3.6 (range 18 to 29) years old, multiracial (White 43%; Hispanic 29%; Other 28%), all living with parents or guardians.
Garcia et al., 2023 [61]	To develop and evaluate participant acceptability and the feasibility of recruitment, retention, adherence, and implementation of a nutrition education and culinary skills intervention for young adults with ASD.	USA—Educational intervention study without a control group. 13 participants with level I ASD diagnosed by a physician: 23% female, aged 26.2 + 4.5 years old, multiracial (White 77%), 31% currently employed, 85% lived with family, 77% with additional diagnosed health condition (ADHD; anxiety disorder; mood disorder), 54% presently taking medication, 16% known food allergies.
Nutritional supplementation and restricted diets as therapeutic strategies in adults with ASD		
Adams et al., 2022 [62]	To evaluate the safety and efficacy of ANRC-Essentials Plus (ANRC-EP), a vitamin/mineral/micronutrient supplement, in children and adults with ASD.	USA—Retrospective survey. 161 participants with a formal diagnosis of ASD: 17 participants over 21 years and 31 (age 16–20). Autism severity: Mild 39 (24%), Moderate 76 (47%), Severe 45 (28%)
Lundbergh et al., 2022 [63]	To explore the effects of *n*-3 long-chain PUFA on cognitive functions in adults with ASD and to determine if these effects are modified by comorbid ADHD.	Denmark—2 × 4 week randomized double-blind crossover trial. 26 adults aged 18 to 40, with a self-reported clinical diagnosis of ASD
Geng et al., 2021 [64]	To gather opinions on the effectiveness of specific food blends (IQed nutritional) and nutrients on speech and motor impairments in individuals with communication delays and/or motor dysfunction symptoms.	USA—Cross-sectional survey study. 77 individuals (age 2–70 years), number of adults not specified.
Adams et al., 2018 [65]	To investigate a comprehensive nutritional and dietary intervention to treat children and adults with ASD.	USA—Randomized, controlled, single-blind 12-month study. 67 children and adults with ASD (ages 3–58 years): ASD treatment group: 6 aged 13 to 20 and 3 adults aged > 20; ASD non treatment group: 7 aged 13 to 20 and 3 adults aged > 20 50 non-sibling neurotypical controls: 11 aged 13 to 20 and 5 adults aged > 20
Matthews & Adams, 2023 [66]	To obtain an understanding of the benefits and adverse effects of therapeutic diets for individuals with ASD, as rated by caregivers of children and adults with ASD (and some individuals with ASD).	USA—Cross-sectional observational study. 818 participants and 25% were over 18 years old (202 adults)
Dietary interventions designed to implement a healthy diet for adults with ASD		
Conti et al., 2024 [67]	To develop canteen menus that meet the nutritional and sensory needs of adults with ASD with the aim of reducing food selectivity and improving their health.	Italy—Intervention pilot study. 22 adults with ASD, aged 19 to 48 years, 72.7% males.
Veneruso et al., 2022 [68]	To promote adaptive behavior and social skills, and reduce the severity of symptomatology through a culinary group intervention in which young people with ASD learn to make fresh pasta by hand.	Italy—Pre-post design study. 20 adults diagnosed with ASD aged 15 to 25 years with a mean age of 19.3 years.
Nabors et al., 2021 [69]	To evaluate the structure, implementation, and outcomes of a healthy eating and exercise program for young adults.	USA—Pilot study. 17 young adults with ASD and ID; 6 parents and 10 staff members.
Gustin et al., 2020 [70]	To assess the impact of a six-week cooking course on autistic college students’ cooking skills, frequency of self-prepared meals, and confidence in meal preparation.	USA—Pilot study. 11 college students with ASD enrolled in the Learning Independence for Empowerment (LIFE) Project.
Hubbard et al., 2015 [71]	To evaluate a Smarter Lunchroom intervention over three months at a residential school for students aged 11–22 with intellectual and developmental disabilities (I/DD).	USA—Pilot Intervention study (The quasi-experimental, pre-post design compared five days of dietary data before and after the intervention). 120 students aged 9–22 years. Number of adults not specified.

ASD = autism spectrum disorder; NT = neurotypical; ID = intellectual disability; AQ = autism spectrum quotient; SATQ = Japanese version of the Subthreshold Autism Trait Questionnaire; ADHD = attention deficit hyperactivity disorder; SWEAA = Swedish Eating Assessment for Autism Spectrum Disorders; PUFA = polyunsaturated fatty acids.

**Table 2 nutrients-17-01456-t002:** Summary of studies on anthropometric measurements of autistic adults.

Reference	Prevalence in ASD Cases	Other Results
Eaves & Ho, 2008 [45]	Overweight = 42%	
Hsieh et al., 2014 [46]	Obesity = 29% Overweight = 29% Normoweight = 37% Underweight = 6%	
Croen et al., 2015 [7]	Obesity = 34%	The prevalence of obesity in the control group was 27.0% (*p* < 0.001) Obesity was diagnosed more frequently among autistic females compared to males (38.5% vs. 32.2%). Adults with ASD had a significantly higher prevalence of obesity than controls 1.41 (1.21–1.64). Both men [1.35 (1.13–1.61)] and women [1.59 (1.19–2.13)] had increased risk compared to unaffected controls. OR (99% CI) = odds ratio (confidence interval; adjusted for sex, age, and race/ethnicity)
Jones et al., 2016 [47]	Obesity = 27% Overweight = 18% Normoweight = 10% Underweight = 18% Missing = 27%	The mean number of co-occurring conditions varied by BMI category (*p* = 0.03); obesity was associated with the highest number of conditions. There were no statistically significant differences in the frequency of current psychotropic medication use by BMI category.
Fortuna et al., 2016 [48]	Obesity = 36.9%	Prevalence of obesity was apparently different between age groups: 18–29, 38.8%; 30–39, 43.3%; >40, 27.8%
Flygare Wallén et al., 2018 [49]	Obesity = 10%	Obesity was diagnosed more frequently among autistic females compared to males (14.9% vs. 7.5%). The prevalence of obesity in the control group was 5.3%. Adults with ASD had a significantly higher prevalence of obesity than controls.
Garcia-Pastor et al., 2019 [50]	Obesity = 14.7% Overweight = 47.1%	Overweight + obesity was higher in men (73.9%) than women (36.4%) (*p* < 0.05) Obesity prevalence was higher in men (17.4%) than women (9.1%) (*p* < 0.05) % body fat was higher in women (29.6) than men (22.1); *p* < 0.05.
Sedgewick et al., 2020 [51]	Obesity = 31% Overweight = 25.7% Normoweight = 37.6% Underweight = 5.4%	The prevalences of obesity, overweight, and underweight in the control group were significantly lower (18.5%, 21.5%, and 2.7%, respectively)
Ptomey et al., 2020 [52]	Obesity = 37.8% Overweight = 24.3% Normoweight = 33.3% Underweight = 4.6%	There were no significant differences in BMI status between males and females.
Hand et al., 2020 [13]	Obesity = 14.4%	The prevalence of obesity in the control group was 10.3%. Obesity was diagnosed more frequently in autistic females than in males (17.9% vs. 12.8%). Older adults with ASD had a significantly higher prevalence of obesity than controls 1.4 (1.3–1.6). Both men [1.3 (1.2–1.5)] and women [1.6 (1.4–1.9)] had increased risk compared to non-autistic subjects. OR (99% CI) *=* odds ratio (confidence interval; adjusted for sex, age, race/ethnicity, rural residence, and estimated household income)
Weir et al., 2021 [53]	Obesity = 27.3% Overweight = 26.6% Normoweight = 39.7% Underweight = 6.3% Missing = 2.3%	The prevalences of obesity, overweight, and underweight in the control group were significantly lower (20.4%, 25.3%, and 3%, respectively) Autistic adults were more likely than controls to be obese [1.335 (1.079–1.651)] or underweight [2.050 (1.309–3.210)], as well as less likely to be in healthy weight [0.676 (0.565–0.809)], even adjusting for potential confounders, i.e., sociodemographic factors, tobacco and alcohol use, sleep and physical activity characteristics, and dietary habits.
Thom et al., 2022 [54]	Obesity = 35.2% Overweight = 27.9%	The proportion of obesity was similar in men (35.3%) and women (34.9%) Overweight prevalence was slightly higher in men (28.3%) than women (26.4%)
Shameem et al., 2024 [55]	Obesity = 51.4% Overweight = 25.0% Not overweight = 23.6%	Overweight and obesity generally decreased with increasing ID severity. Obesity was more prevalent in people with no ID (63.8%) than ASD + ID (45.5%)

ASD = autism spectrum disorder; no data about ASD severity; ID = intellectual disability; SD = standard deviation; obesity: BMI ≥ 30 Kg/m^2^; overweight: BMI > 25 kg/m^2^ and ≤30 kg/m^2^; underweight: BMI ≤ 18.5 kg/m^2^.

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
