# Peer review of "Food and Nutrition in Autistic Adults: Knowledge Gaps and Future Perspectives"

_nutrients, 2025, doi:10.3390/nu17091456_

Round 1

Reviewer 1 Report

Comments and Suggestions for Authors

General comments:

Sara Remón et al reviews synthesizes most of the current studies related to nutrition in adults with autism spectrum disorder (ASD).This paper systematically elaborates on existing research regarding: the impact of understanding food preferences and sensory processing on ASD adults' nutritional status, methods for assessing nutritional conditions in ASD adults, and current dietary/nutritional interventions affecting their eating behaviors. The article holds significant value for establishing standardized dietary guidelines for ASD adults in the future. While the content demonstrates clear logical structure and incorporates critical analysis of research limitations, certain details still require further refinement.

Specific comments:

  1. Some claims lacks of citations .For instance, the sentence “the prevalence of which varies depending on the age group” (lines 48–49) requires supporting references to validate this assertion.
  2. Inconsistent abbreviation usage.The abbreviation“AS” is introduced without prior definition (lines 57, 71, 90). Given the context, it likely refers to “ASD” (Autism Spectrum Disorder).
  3. Grammatical inconsistency.In line 226, clarify whether“affect” or “affected” is contextually appropriate based on tense and sentence structure.
  4. Some sentences are incomplete.For example, the term“abilities” (lines 238–239) lacks specificity. Clarify whether it refers to sensory sensitivity, food identification skills, or other domains.
  5. Some parts of the elaboration lack a subject.The sentence“These preliminary studies show dislikes for foods with particular textures and strong tastes” needs explicit attribution. Revise to: “These preliminary studies suggest that ASD adults exhibit dislikes for foods with particular textures and strong tastes.”
  6. Some expressions need to be rephrased to support the relevant arguments.The statement“Studies on fatty acid supplementation regarding cognitive and behavioral aspects in adults with ASD seem particularly interesting” could be strengthened. Consider: “Research exploring the effects of fatty acid supplementation on cognitive and behavioral outcomes in ASD adults holds significant theoretical and clinical relevance.”

Author Response

Response to Reviewer 1 Comments

Thank you very much for taking the time to review this manuscript. Please find the detailed responses below and the corresponding revisions/corrections in track changes in the re-submitted files

Point-by-point response to Comments and Suggestions for Authors

●      General comments: Sara Remón et al. reviews synthesizes most of the current studies related to nutrition in adults with autism spectrum disorder (ASD). This paper systematically elaborates on existing research regarding: the impact of understanding food preferences and sensory processing on ASD adults' nutritional status, methods for assessing nutritional conditions in ASD adults, and current dietary/nutritional interventions affecting their eating behaviors. The article holds significant value for establishing standardized dietary guidelines for ASD adults in the future. While the content demonstrates clear logical structure and incorporates critical analysis of research limitations, certain details still require further refinement.

Response: We thank the reviewer for their thoughtful and encouraging feedback. We are pleased that the structure and focus of the article were clear and that the relevance of the topic was recognized. We now address the specific details mentioned in your review.

●      Comments 1:  Some claims lacks of citations. For instance, the sentence “the prevalence of which varies depending on the age group” (lines 48–49) requires supporting references to validate this assertion.

Response 1: Thank you for pointing this out. We agree with this comment. However, the new wording of this section no longer contains the sentence that needed to be adequately referenced.

●      Comments 2:  Inconsistent abbreviation usage. The abbreviation “AS” is introduced without prior definition (lines 57, 71, 90). Given the context, it likely refers to “ASD” (Autism Spectrum Disorder).

Response 2: Agree. We have, accordingly, revised the whole document to correct this point. The abbreviation ASD is now consistently used throughout the manuscript.

●      Comments 3:  Grammatical inconsistency. In line 226, clarify whether “affect” or “affected” is contextually appropriate based on tense and sentence structure.

Response 3: Thank you for pointing this out. We agree with this comment. Therefore, we have reworded the sentence in this way: “Atypical sensory experiences are reported to occur in as many as 90% of individuals with ASD. They may affect almost every sensory modality: vision, audition, smell, touch, and taste” (lines 278-280)

●      Comments 4:  Some sentences are incomplete. For example, the term “abilities” (lines 238–239) lacks specificity. Clarify whether it refers to sensory sensitivity, food identification skills, or other domains.

Response 4: Thank you for pointing this out. We agree with this comment. However, the new wording of this section no longer contains the sentence that needed to be completed.

●      Comments 5:  Some parts of the elaboration lack a subject. The sentence “These preliminary studies show dislikes for foods with particular textures and strong tastes” needs explicit attribution. Revise to: “These preliminary studies suggest that ASD adults exhibit dislikes for foods with particular textures and strong tastes.”

Response 5: Thank you for your suggestion. The sentence has been revised to read: “These preliminary studies suggest that ASD adults exhibit dislikes for foods with particular textures and strong tastes.” (lines 389-391)

●      Comments 6:  Some expressions need to be rephrased to support the relevant arguments. The statement “Studies on fatty acid supplementation regarding cognitive and behavioral aspects in adults with ASD seem particularly interesting” could be strengthened. Consider: “Research exploring the effects of fatty acid supplementation on cognitive and behavioral outcomes in ASD adults holds significant theoretical and clinical relevance.”

Response 6: Thank you for your suggestion. The sentence has been revised to read: “Research exploring the effects of fatty acid supplementation on cognitive and behavioral outcomes in adults with ASD holds significant theoretical and clinical relevance.” (lines 858-860)

Additional clarifications

In response to one of the reviewer’s feedback, we have implemented several modifications to our manuscript. These changes have primarily focused on clarifying the scope of our review, which specifically targets adults with Autism Spectrum Disorder (ASD). The modifications have allowed us to enhance the manuscript's clarity, reduce length in certain sections, and prevent potential misunderstandings about the review's objectives, while maintaining the integrity of our original content and conclusions.

Key Modifications:

●      Abstract: We have incorporated a clear statement regarding the review's objective: "It is therefore essential to critically review existing research focusing on autistic adults to draw robust conclusions and identify clear research gaps."

●      Introduction:

o Removed information not directly relevant to the introduction and justification of our review

o Restructured text to improve clarity

o Consolidated information about children with ASD into concise paragraphs

●      Methodology: It explicitly emphasize our consistent focus on adult populations by highlighting "adult/s" as a key search term and it reinforce that our inclusion criteria specifically targeted studies with adult ASD populations

●      Section 3: Summarized data on autistic children into two focused paragraphs

●      Section 4: Consolidated sub-sections 4.2 and 4.3 into a single, more cohesive section titled "Biochemical and clinical assessment of nutritional status in adults with ASD".

We believe that the changes derived from the revisions have strengthened the manuscript's methodological rigor and improved its overall focus on the adult ASD population, addressing the primary concerns raised during the review process. The core scientific content and conclusions remain unchanged, as they were correctly aligned with our stated objectives from the outset

Reviewer 2 Report

Comments and Suggestions for Authors

As much of the research on the prevalence, origins, and consequences of eating problems in autism comes from studies of children with ASD, there is little information relevant to adults. This literature review searched the electronic databases Science Direct, PubMed, Scopus, Web of Science, and Google Scholar for studies published between 2013 and May 2024 that analyzed food selectivity in adults with autism, highlighting a clear trend toward selective eating, largely influenced by food perception and attempts to find a promising intervention strategy combining sensory adaptation and structured meal planning for adults with ASD. Overall, this is a good topic and research point, but there are still several issues that need to be clarified and identified by the authors. 

1. However, this article is too long and has a tendency to be overly wordy. It would have been better to include some comparative tables to facilitate the presentation of the differences in food preferences and sensory processing, nutritional status, and dietary and nutritional interventions between children and adults with ASD so that the reader could easily understand and follow. 

2. In terms of summarizing dietary and nutritional intervention studies as well, it would be useful for the authors to provide a table comparing different dietary interventions for children and adults with ASD. 

3. What's “ID”, “AS”? Please provide the full name/description of the abbreviation when it first appears. 

Author Response

Response to Reviewer 2 Comments

Thank you very much for taking the time to review this manuscript. Please find the detailed responses below and the corresponding revisions/corrections in track changes in the re-submitted files.

Point-by-point response to Comments and Suggestions for Authors

●      General comments: As much of the research on the prevalence, origins, and consequences of eating problems in autism comes from studies of children with ASD, there is little information relevant to adults. This literature review searched the electronic databases Science Direct, PubMed, Scopus, Web of Science, and Google Scholar for studies published between 2013 and May 2024 that analyzed food selectivity in adults with autism, highlighting a clear trend toward selective eating, largely influenced by food perception and attempts to find a promising intervention strategy combining sensory adaptation and structured meal planning for adults with ASD. Overall, this is a good topic and research point, but there are still several issues that need to be clarified and identified by the authors.

Response: We sincerely thank the reviewer for their thoughtful feedback and for recognizing the relevance and potential of the topic. We appreciate your interest in our work and agree that further clarification and refinement are needed in some areas. In response, we have carefully addressed each of the points raised and revised the manuscript accordingly to strengthen the scientific rigor and clarity of our review

●      Comments 1: However, this article is too long and has a tendency to be overly wordy. It would have been better to include some comparative tables to facilitate the presentation of the differences in food preferences and sensory processing, nutritional status, and dietary and nutritional interventions between children and adults with ASD so that the reader could easily understand and follow.

●      Comments 2: In terms of summarizing dietary and nutritional intervention studies as well, it would be useful for the authors to provide a table comparing different dietary interventions for children and adults with ASD.

Response 1 & 2: We appreciate the reviewer’s observations. We fully agree that a comprehensive comparison between children and adults would be highly valuable; however, such an analysis falls outside the scope of the present review and would considerably extend its length—this is acknowledged as one of the limitations of our manuscript. We consider this a meaningful direction for future work and potentially the focus of a follow-up publication, once conclusions regarding the adult population are firmly established.

During the exhaustive review of the manuscript to incorporate your valuable suggestions, we have identified that including data from children could lead to confusion, as the main objective of this article is to explore and better understand Food and Nutrition in Autistic Adults. To improve clarity and focus, we have addressed several modifications, most of them aim to remove any potential ambiguity regarding our study's focus on adults with ASD:

Abstract: A clarification on the objective of the review is incorporated: “It is therefore essential to critically review existing research focusing on autistic adults to draw robust conclusions and identify clear research gaps.” [Lines 23-24]

Introduction:

•  Information not relevant to the introduction and justification of the objective of the review has been removed [Line 59; Lines 114-123]

•  In order to provide greater clarity in content, some of the text has been rearranged [Lines 123-129 moved to Lines 60-66]

•  The specific data on autistic children have been summarized in one paragraph: “Consequently, some nutritional concerns have been identified such as weight problems, and various nutritional deficiencies.” [Lines 79-81]

•  There have been no changes in the bibliographic references.

Methodology: We would like to clarify that, as previously stated in our methodology section, our only obligatory inclusion criterion was that the study included an adult population with ASD. To strengthen the methodological rigor of our approach and address your concern, we have explicitly emphasized that our search strategy and subsequent analysis consistently targeted adult populations by incorporating the term "adult/s" as one of the search keywords, which accurately reflects our implemented methodology.

Results:

•  Section 3: The specific data on autistic children have been summarized in two paragraphs:

o   “Numerous studies have focused on FS in children with ASD, demonstrating that FS is more common in this population than in typically developing children, with rates as high as 85%. This limited food repertoire is linked to nutrient inadequacies, and research suggests that FS may persist into adulthood if left untreated.” [Lines 210-214]

o   “Research results on sensory processing in autistic population, mainly children, show inconsistent findings across odor detection and identification, visual processing of food stimuli, and textural perception of food in the mouth, but overall suggests that atypical sensory responses are closely linked to food refusal behaviors.” [Lines 284-289]

•  Section 4: Two main changes have been made:

o   We have consolidated sections 4.2 and 4.3 into a single section entitled Biochemical and clinical assessment of nutritional status in adults with ASD, as the findings from these two sections can be concluded in a similar manner. So, a new paragraph has been included in lines 569-573: “These biochemical and clinical assessment results indicate significant knowledge gaps regarding potential nutritional deficiencies in adults. Current findings suggest that many aspects of adult nutritional status remain poorly understood, necessitating further comprehensive research to establish a complete picture in this regard.”

o   Besides, the paragraph in lines 671-678 has been reworded: “Based on our comprehensive review, we cannot establish whether nutrient intake inadequacies are present in formally diagnosed adults with ASD, as none of the reviewed studies provide information in this regard.”

•  Section 5: We have reviewed this section, and as authors, we believe that no adaptations are required to remove any potential ambiguity regarding our study's focus on adults with ASD.

●      Comments 3:  What's “ID”, “AS”? Please provide the full name/description of the abbreviation when it first appears.

Response 3:

We appreciate the reviewer’s observation.

Regarding ID, it means intellectual disability (line 52).

Certainly, the use of “AS” was incorrect, and the proper abbreviation is “ASD” (Autism Spectrum Disorder). This has been corrected throughout the manuscript.

Additionally, we have carefully reviewed the entire text to ensure that all abbreviations are correctly defined at their first appearance and used consistently.

Reviewer 3 Report

Comments and Suggestions for Authors

The vulnerable population eating disorders are difficult to manage given the difficulty to communicate with such persons. When the persons are member of a group with potential psychological disorders it is even more difficult and chalangeable. I might say that such persons are treated more or less like children, while their bodies evolve as adults. They are very selective with the food, either refuse many dishes or overeat. Most information related to food disorders or eating disorders for ASD is based on studies focused on autistic children while this should not be applied to autistic adults. This topic is very interesting from this point of view. Few articles are published on this field. The current proposed review discovered 43 full-text articles for a period of 11 years (2013-2024) by searching specific databases (PubMed, Science Direct, Scopus, Web of Science). The only including criteria was the the study was focused on adult population with ASD. Reviews, commentaries, editorials and opinions were excluded. Only original research publications were taken into account. The information on the chosen studies is shown on a table based format structured on various topics such as food selectivity in autistic adults, sensory processing of food, anthropometric measurements and so on. Each of this topics was discussed in subsections. Actually, the article is well organized in sections and subsections making it easy to follow.
The conclusion are supposed by the whole text. The identified trend on food selectivity in autistic adults is that such disorder is influenced by sensory perception. The taste, the texture, smell and temperature are driven the nutritional choices for autistic adults. Preferences to certain food and aversion to others are pushed to the extreme and might lead to unbalanced diet affecting the whole health. Future studies should be focused on the anthropometric measurements such as waist circumference, waist-to-hip ration, waist-to-height ration. This would align with the new obesity framework while addressing specific dietary needs.
The paper is based on a large reference list that supports the topic.

Author Response

Response to Reviewer 3 Comments

Thank you very much for taking the time to review this manuscript. Please find the detailed responses below and the corresponding revisions/corrections in track changes in the re-submitted files.

Point-by-point response to Comments and Suggestions for Authors

Comments: The vulnerable population eating disorders are difficult to manage given the difficulty to communicate with such persons. When the persons are member of a group with potential psychological disorders it is even more difficult and chalangeable. I might say that such persons are treated more or less like children, while their bodies evolve as adults. They are very selective with the food, either refuse many dishes or overeat. Most information related to food disorders or eating disorders for ASD is based on studies focused on autistic children while this should not be applied to autistic adults. This topic is very interesting from this point of view. Few articles are published on this field. The current proposed review discovered 43 full-text articles for a period of 11 years (2013-2024) by searching specific databases (PubMed, Science Direct, Scopus, Web of Science). The only including criteria was the the study was focused on adult population with ASD. Reviews, commentaries, editorials and opinions were excluded. Only original research publications were taken into account. The information on the chosen studies is shown on a table based format structured on various topics such as food selectivity in autistic adults, sensory processing of food, anthropometric measurements and so on. Each of this topics was discussed in subsections. Actually, the article is well organized in sections and subsections making it easy to follow.

The conclusion are supposed by the whole text. The identified trend on food selectivity in autistic adults is that such disorder is influenced by sensory perception. The taste, the texture, smell and temperature are driven the nutritional choices for autistic adults. Preferences to certain food and aversion to others are pushed to the extreme and might lead to unbalanced diet affecting the whole health. Future studies should be focused on the anthropometric measurements such as waist circumference, waist-to-hip ration, waist-to-height ration. This would align with the new obesity framework while addressing specific dietary needs.

The paper is based on a large reference list that supports the topic.

Response:

We would like to thank the reviewer for their time and comprehensive summary of our manuscript. We greatly appreciate their thoughtful and encouraging feedback. We are pleased that the relevance of the topic was recognized. We are also grateful that you have acknowledged the well-organized structure of our paper and that our conclusions are supported by the content presented.

Your comments motivate us to continue working in this line of research with the aim of understanding how to improve the diet and nutritional status of adults with ASD and, consequently, their health."

Additional clarifications

 In response to one of the reviewer’s feedback, we have implemented several modifications to our manuscript. These changes have primarily focused on clarifying the scope of our review, which specifically targets adults with Autism Spectrum Disorder (ASD). The modifications have allowed us to enhance the manuscript's clarity, reduce length in certain sections, and prevent potential misunderstandings about the review's objectives, while maintaining the integrity of our original content and conclusions.

Key Modifications:

●      Abstract: We have incorporated a clear statement regarding the review's objective: "It is therefore essential to critically review existing research focusing on autistic adults to draw robust conclusions and identify clear research gaps."

●      Introduction:

o Removed information not directly relevant to the introduction and justification of our review

o Restructured text to improve clarity

o Consolidated information about children with ASD into concise paragraphs

●      Methodology: It explicitly emphasize our consistent focus on adult populations by highlighting "adult/s" as a key search term and it reinforce that our inclusion criteria specifically targeted studies with adult ASD populations

●      Section 3: Summarized data on autistic children into two focused paragraphs

●      Section 4: Consolidated sub-sections 4.2 and 4.3 into a single, more cohesive section titled "Biochemical and clinical assessment of nutritional status in adults with ASD".

We believe that the changes derived from the revisions have strengthened the manuscript's methodological rigor and improved its overall focus on the adult ASD population, addressing the primary concerns raised during the review process. The core scientific content and conclusions remain unchanged, as they were correctly aligned with our stated objectives from the outset.

Round 2

Reviewer 2 Report

Comments and Suggestions for Authors

Compared to the previous manuscript, this one is a significant improvement in terms of writing. The authors responded appropriately to most of the questions posed by the reviewer. However, the reviewer still felt that it would have been useful for the authors to provide a tabular data comparing the effects of different dietary interventions for children and adults with ASD to facilitate readers to follow up and further understand the focus and value of this study.

Author Response

Response to Reviewer 2 Comments

Thank you very much for taking the time to review this revised version of our manuscript. Please find the detailed response below and the corresponding revisions/corrections highlighted in the re-submitted file.

Point-by-point response to Comments and Suggestions for Authors

General comments: Compared to the previous manuscript, this one is a significant improvement in terms of writing. The authors responded appropriately to most of the questions posed by the reviewer. However, the reviewer still felt that it would have been useful for the authors to provide a tabular data comparing the effects of different dietary interventions for children and adults with ASD to facilitate readers to follow up and further understand the focus and value of this study.

Response: We greatly appreciate the reviewer’s thoughtful suggestion to include a comparative table summarizing the effects of dietary interventions in children and adults with ASD. However, we have decided not to include such a table in the current version of the manuscript, as it would considerably extend the discussion and shift the focus away from our main objective, which is not to compare pediatric and adult populations.

That said, we fully recognize the value of this perspective. In response to the reviewer’s comment, we have revised the section on dietary interventions and added a paragraph that briefly contextualizes this body of research (lines 581-597), citing the most relevant reviews conducted in children (new references 103 to 114 have been added). This addition aims to guide interested readers toward existing literature while maintaining the focus of our study.

We believe that a detailed comparison between dietary intervention outcomes in children and adults with ASD represents a valuable direction for future research that merits dedicated attention in subsequent publications
